



# A new method for post-processing daily sub-seasonal to seasonal rainfall forecasts from GCMs and evaluation for 12 Australian catchments

Andrew Schepen[1], Tongtiegang Zhao[2], Quan. J. Wang[3] and David E. Robertson[2]

[1]CSIRO Land and Water, Dutton Park, 4102, Australia
[2]CSIRO Land and Water, Clayton, 3168, Australia
[3]Department of Infrastructure Engineering, The University of Melbourne, Parkville, 3010, Australia

*Correspondence to*: Andrew Schepen (Andrew.Schepen@csiro.au)

**Abstract.** Rainfall forecasts are an integral part of hydrological forecasting systems at sub-seasonal to seasonal time scales. In
seasonal forecasting, global climate models (GCMs) are now the go-to source for rainfall forecasts. However, for hydrological
applications, GCM forecasts are often biased and unreliable in uncertainty spread, and therefore calibration is required before
use. There are sophisticated statistical techniques for calibrating monthly and seasonal aggregations of the forecasts. However,
calibration of seasonal forecasts at the daily time step typically uses very simple statistical methods or climate analogue
methods. These methods generally lack the sophistication to achieve unbiased, reliable and coherent forecasts of daily amounts
and seasonal accumulated totals. In this study, we propose and evaluate a Rainfall Post-Processing method for Seasonal
forecasts (RPP-S) based on the Bayesian joint probability approach for calibrating daily forecasts and the Schaake Shuffle
approach for connecting the daily ensemble members of different lead times. We apply the method to post-process ACCESS-
S forecasts for 12 perennial and ephemeral catchments across Australia and for 12 initialisation dates. RPP-S significantly
reduces bias in raw forecasts and improves both skill and reliability. RPP-S forecasts are more skilful and reliable than forecasts
derived from ACCESS-S forecasts that have been post-processed using quantile mapping, especially for monthly and seasonal
accumulations. Several opportunities to improve the robustness and skill of RPP-S are identified. The new RPP-S post-
processed forecasts will be used in ensemble sub-seasonal to seasonal streamflow applications.

## 1 Introduction

Rainfall forecasts are an integral part of hydrological forecasting systems at sub-seasonal to seasonal time scales (Crochemore
et al., 2017;Bennett et al., 2016;Wang et al., 2011). Inclusion of climate information in seasonal streamflow forecasts enhances
streamflow predictability (Wood et al., 2016). One strategy for integrating climate information into hydrological models is to
conditionally resample historical rainfall (e.g. Beckers et al., 2016;Wang et al., 2011). An alternative approach is to use rainfall
forecasts from dynamical climate models.



Ensemble rainfall forecasts from GCMs (global climate models) are attractive for hydrological prediction in that they forecast multiple seasons ahead and have a well-established spatial and temporal forecast structure. On the other hand, a major issue with GCM forecasts at sub-seasonal to seasonal time scales is that the forecasts are often biased and lacking in predictability of local climate (e.g.Kim et al., 2012;Tian et al., 2017). It is therefore necessary to post-process GCM rainfall forecasts using

statistical or dynamical methods before they can be used in hydrological models (Yuan et al., 2015).

Several conceptually simple statistical correction methods are used for directly post-processing daily GCM rainfall forecasts including: additive bias correction, multiplicative bias correction and quantile mapping (Ines and Hansen, 2006). For example, Crochemore et al. (2016) recently evaluated linear scaling and quantile mapping for post-processing ECMWF System4 rainfall forecasts in France. Quantile mapping adjusts forecast means and ensemble spread but it is not a full calibration method

because it does not account for the correlation between forecasts and observations (Zhao et al., 2017). It is useful for bias correction of climate change projections where a full statistical calibration is inappropriate (Teutschbein and Seibert, 2012). Since additive and multiplicative bias correction and quantile mapping methods do not account for intrinsic GCM skill they are ineffective to use as post-processing tools when GCM forecasts are unskilful.

Post-processing methods that take into account model skill typically fall under the model output statistics (MOS; Glahn and

Lowry, 1972) banner. MOS type approaches are well established in the weather forecasting community for short term forecasting and modern variants are normally probabilistic. For example Wilks and Hamill (2007) studied ensemble MOS approaches for post-processing global forecast system (GFS) forecasts of rainfall and temperature up to 14 days ahead.

MOS methods can also be thought of as full calibration methods. In this regard, several Bayesian calibration approaches are known to be effective at post-processing GCM rainfall totals aggregated to monthly and seasonal time scales (Hawthorne et

al., 2013;Schepen et al., 2014;Luo et al., 2007). However, it is apparent that full calibration methods aren't normally applied to the post-processing of daily GCM forecasts of rainfall in the sub-seasonal to seasonal period.

That said, some studies have explored more sophisticated methods for post-processing daily rainfall forecasts from GCMs. For example, Pineda and Willems (2016) applied a non-homogenous hidden Markov model (NHHMM) to forecasts in the northwestern region of South America. Their method extracted information from GCM forecasts of rainfall fields and SSTs.

Their study built on the NHHMM method originally developed by Robertson et al. (2004) for prediction of rainfall occurrence in Brazil. In Australia, Shao and Li (2013) and Charles et al. (2013) applied an analogue downscaling method (Timbal and McAvaney, 2001) to produce downscaled daily rainfall forecasts from POAMA (the Predictive Ocean Atmosphere Model for Australia). The NHHMM and analogue methods are not straightforward to apply operationally, as they require the identification of optimal climate predictors in different climatic regions. The methods do not by design lead to forecasts that

are always reliable in ensemble spread and at least not worse than climatology forecasts.

Statistical post-processing of daily rainfall forecasts is a formidable challenge, perhaps explaining the lack of sophisticated methods. Barriers include short GCM hindcast records, a high prevalence of zero rainfall amounts, seasonal variations in rainfall patterns, and intrinsically low GCM skill. Amplifying the challenge is that GCM skill decays rapidly as the age of ensemble members increases. For example, Lavers et al. (2009) examined temperature and rainfall forecasts using DEMETER





and CFS GCMs, and found that, in an idealized scenario, skill in the first 30 days is primarily attributable to skill in the first 15 days and much less to skill over the next 15 days. Post-processing methods should therefore be designed to capture as much skill as possible in the first fortnight and take into account GCM skill when post-processing forecasts further ahead.

In this study, we seek to develop a new, more direct daily rainfall post-processing method that operates solely on rainfall output

from GCMs and provides a full forecast calibration taking into account GCM skill. We build a new sub-seasonal to seasonal rainfall post-processor, which has some similarities to the rainfall forecast post-processor (RPP) developed by Robertson et al. (2013b) and Shrestha et al. (2015) for post-processing numerical weather prediction (NWP) forecasts for short-term streamflow forecasting. The new system is hereafter called RPP-S, which stands for rainfall forecast post-processor – seasonal.

The proposed RPP-S method applies the Bayesian joint probability (BJP) modelling approach to post-process daily GCM

forecasts of rainfall. BJP has never before been used in this situation and it is therefore important to fully evaluate the merits of BJP as a component of a daily forecast post-processing system. As GCMs can produce ensemble forecasts for over 100 days ahead, RPP-S is developed to generate daily rainfall amounts that aggregate to intra-seasonal and seasonal totals. To this end, the Schaake Shuffle (Clark et al., 2004) is also included as a component of RPP-S.

We apply RPP-S to post-process raw catchment rainfall forecasts from ACCESS-S, Australia's new seasonal forecasting

GCM. We comprehensively assess bias, skill and reliability for 12 ephemeral and perennial catchments across Australia. RPP-S catchment rainfall forecasts are compared to catchment rainfall forecasts derived from the Bureau of Meteorology's bias-corrected product for ACCESS-S. Opportunities to develop the method further are discussed.

## 2 Data and catchments

### 2.1 ACCESS-S rainfall forecasts

Raw, gridded GCM rainfall forecasts are obtained from the BoM's new ACCESS-S forecasting system. Raw catchment rainfall forecasts are derived through a process of area-weighted averaging of the ensemble mean.

ACCESS-S is a customised version of the UK Met Office's seasonal forecasting system. It contains a fully coupled model representing the interactions among the Earth's atmosphere, oceans, land surface and sea ice. The current horizontal spatial resolution of the ACCESS-S atmospheric model is approximately 60 km in the mid-latitudes. Full details of the current system

are provided by Hudson et al. (2017).

ACCESS-S hindcasts are initialised at midnight UTC on days 1 and 25 of each month. A burst ensemble comprising 11 ensemble members is generated. Hindcasts are available for the period 1990–2012 (23 years). A longer hindcast set is not possible because of a lack of suitable initial conditions. The next version of ACCESS-S will be delivered with more sets of initial conditions and longer hindcasts.

The BoM post-processes ACCESS-S forecasts using quantile mapping (QM) at the 60 km grid scale. We use QM forecasts as a benchmark for RPP-S forecasts. QM catchment rainfall forecasts are derived in the same way as for raw catchment rainfall forecasts.



## 2.2 Observed rainfall

Observed rainfall is derived from the Australian Bureau of Meteorology's 5km daily rainfall analysis (AWAP). We make use of AWAP data from 1950 onwards (further details in section 3.2.3). Catchment rainfall is derived through a process of area-weighted averaging.

An important note on time zones: Australian rainfall is recorded as 24 hour totals to 9 am local time. Consequently, AWAP data are not perfectly synchronised with the ACCESS-S forecasts, which are initialised at midnight UTC. The asynchronism is compounded by time zone differences and daylight savings.

We align the GCM data and observed data as best we can. For east coast locations, an ACCESS-S forecast overlaps with the following day's AWAP rainfall analysis with a discrepancy of 1-2 hours. The time discrepancy for west coast locations is
approximately 2 hours more.

## 2.3 Catchments

Twelve perennial and ephemeral catchments spread over Australia are selected for application and evaluation. Catchment information including name, gauge ID, regional location and size are shown in Table 1. Catchment locations and climate zones are mapped in Figure 1. The catchments reside in highly distinct climate zones and vary markedly in size from 100 km$^2$ to
119034 km$^2$. Evaluation across climate zones and for varying catchment sizes helps draw more general conclusions about post-processing effectiveness. However, relating forecast performance to catchment characteristics is outside the scope of this study.

## 3 Methods

### 3.1 Bayesian joint probability models

Our post-processing method embeds the Bayesian joint probability (BJP) modelling approach, which was originally designed
for forecasting seasonal streamflow totals (Wang and Robertson, 2011;Wang et al., 2009;Robertson et al., 2013a). BJP has since been applied to calibrate hourly rainfall forecasts (Shrestha et al., 2015;Robertson et al., 2013b) and seasonal rainfall forecasts (Hawthorne et al., 2013;Khan et al., 2015;Peng et al., 2014;Schepen and Wang, 2014;Schepen et al., 2014). BJP was most recently adapted for sub-seasonal to seasonal streamflow forecasting (Zhao et al., 2016;Schepen et al., 2016). BJP has never been applied to post-process daily rainfall forecasts from GCMs.
Salient details of the BJP modelling approach are provided here. Full details can be found by reference to the above studies. How the models are used in RPP-S is described in the next section.

BJP formulates a joint probability distribution to characterize the relationship between forecast ensemble means (predictors) and corresponding observations (predictands). The joint distribution is modelled as a bivariate normal distribution after transformation of the marginal distributions. Data transformation is handled using the flexible log-sinh transformation (Wang
et al., 2012). The log-sinh transformation transforms $y$ by




$$f(y) = \beta^{-1} \ln(\sinh(\alpha + \beta y))  \tag{1}$$

where $\alpha$ and $\beta$ are transformation parameters. A predictor $x$ is transformed to $g$. Likewise, a predictand $y$ is transformed to $h$. Therefore

$$\begin{bmatrix} g \\ h \end{bmatrix} \sim N(\boldsymbol{\mu}, \boldsymbol{\Sigma})  \tag{2}$$

where

$$\boldsymbol{\mu} = \begin{bmatrix} \mu_g \\ \mu_h \end{bmatrix}  \tag{3}$$

is the vector of means and

$$\boldsymbol{\Sigma} = \begin{bmatrix} \sigma_g^2 & \rho_{gh}\sigma_h\sigma_g \\ \rho_{gh}\sigma_h\sigma_g & \sigma_h^2 \end{bmatrix}  \tag{4}$$

is the covariance matrix.

In fitting mode, BJP model parameters are inferred from data. The transformation parameters is estimated by maximum a posteriori probability (MAP) solution. The parameters of the bivariate normal distribution are inferred from a Bayesian formulation, using a Markov chain Monte Carlo (MCMC) algorithm to sample 1000 sets of parameters to represent parameter uncertainty.

In forecasting mode, a BJP model is conditioned on new data. For a particular parameter set, $\theta$, and new transformed predictor

value $g_{new}$,

$$h_{new} \mid g_{new}, \theta \sim N\left( \mu_h + \rho_{gh}\frac{\sigma_h}{\sigma_g}(g_{new} - \mu_g), \sigma_h^2(1 - \rho_{gh}^2) \right)  \tag{5}$$

A sample of $y_{new}$ is obtained by drawing a random sample of $h_{new}$ from Eq. (5) and then back-transforming using the inverse

of Eq. (1). A complete forecast ensemble is built up by sampling a single $y_{new}$ for each parameter set.

We note some specifics of the BJP implementation used in this study. Values of zero rainfall are treated as censored values,

meaning the true value is assumed to be less than or equal to zero. The predictor and predictand censor thresholds are $x_c = 0$

and $y_c = 0$, respectively. Data are rescaled to lie in the range [0,5] prior to transformation to simplify the inference of

transformation parameters. To prevent undesirable extrapolation effects in forecasting mode, predictor values are limited to





twice the maximum predictor value used to fit the model. The strategy of limiting predictor values has been used successfully in earlier studies (e.g. McInerney et al., 2017; Wang and Robertson, 2011).

## 3.2 The seasonal rainfall forecast post-processor (RPP-S)

### 3.2.1 Pooling of data

Model parameters may be poorly estimated when there are insufficient nonzero data values used in data inference. In very dry catchments, certain days of year may have forecasts or observations with no non-zero values at all. In such cases, inference is not possible. To overcome this problem, predictor and predictand data sets are created by pooling data for consecutive days and for multiple GCM initialisation dates. Figure 2 illustrates our RPP-S configuration using 6 BJP models to cover 112 days. The RPP-S method places no restriction on the configuration, and alternative configurations are discussed in section 5. We

pool daily data for six "day groups": week 1, week 2, weeks 3–4, and subsequent 4 week periods. The data in day group 1 are used to fit model BJP1, the data in day group 2 are used to fit model BJP2, and so on. The size of the day groups expands farther from the initialisation date. Smaller day groups in early periods are intended to extract skill from the initial conditions. Larger day groups in later periods are intended to better approximate the climatological distribution.

We also pool data for multiple GCM initialisation dates. As noted in section 2.1, ACCESS-S hindcasts are initialised on days

1 and 25 of each month. We assume it is better to estimate the correlation for a day group rather than for each day separately (see section 5 for further discussion). Predictor and predictand data are paired according to the day since forecast initialisation. For example, in Figure 2, days 1–7 from the 1 February and 25 January runs are pooled and used to fit model BJP1.

Pooled data are not completely independent meaning there is potential to underestimate parameter uncertainty using BJP. Nevertheless, we expect the effect to be limited by the weak predictor-predictand relationships and weak persistence in daily

rainfall. Additionally, we have not dealt with overlapping data in any special way (see section 5 for further discussion).

### 3.2.2 Fitting BJP models

After pooling the data, model fitting for each BJP model proceeds as follows:

   (1)  Rescale predictor and predictand data so that each series ranges within [0,5]

   (2)  Estimate the log-sinh transformation parameters for the predictors using maximum a posteriori (MAP) solution

(3)  Estimate the log-sinh transformation parameters for the predictands in the same way

   (4)  Apply the transformations to normalise the predictor and predictand data

   (5)  Apply the transformations to transform the predictor and predictand censoring thresholds

   (6)  Sample parameter sets representing the posterior distribution of the bivariate normal distribution parameters using MCMC sampling, the transformed data and transformed censoring thresholds.





### 3.2.3 Forecasting

A BJP model is used to forecast all of the days in the group. For example, model BJP1 is used to forecast days 1–7. Forecasting a single day using BJP proceeds as follows:

    (1) Transform the predictor value using the log-sinh transformation for predictors

(2) Sample one $h_{new}$ for each parameter set

    (3) Back-transform the ensemble members using the transformation for predictands

    (4) Rescale the ensemble members to the original space (opposite of step (1) in 3.2.2).

    (5) Set negative values to zero

BJP forecast ensemble members are randomised and are not linked across days by default. To deal with the problem, we apply the Schaake Shuffle (Clark et al., 2004), which uses historical data to link ensemble members and create sequences with realistic temporal patterns. Fifty years of historical data from 1950 is used to apply the Schaake Shuffle, thus the forecast ensemble shuffled in blocks of 50.

An example RPP-S forecast for the BRS catchment is shown in Figure 3. The top panel is the forecast of daily amounts and
the bottom panel is the forecast of accumulated totals. Hereafter, a daily amount is taken to mean a 24-hour rainfall total on any given day. An accumulated total is the sum of daily rainfall amounts over a number of days. In this example, the forecast correctly predicts a dry beginning to the forecast period. The quantile ranges of the daily amounts are reasonably consistent with the observed values. The accumulated forecast is somewhat narrower than the climatology reference forecast at monthly and seasonal time scales and the forecast is predicting a seasonal rainfall total less than the climatological median. It is clear
that RPP-S is able to produce ensemble forecasts of daily rainfall amounts.

### 3.3. Verification

We use RPP-S to post-process all available ACCESS-S forecasts initialised on day 1 of each month for 100 days ahead (the sub-seasonal to seasonal forecast period). For water resources management, whilst it is important that daily amounts are realistic, it is vital that the accumulated totals are as reliable and as skilful as possible. Forecasts are verified against both daily
amounts and accumulated totals.

RPP-S forecasts are generated using leave-one-year-out cross-validation. RPP-S forecasts are compared with the BoM's QM forecasts (see section 2.1), which are also generated using leave-one-year-out cross-validation.

Bias is recognised as the correspondence between the mean of forecasts and the mean of observations. Bias is visually assessed by plotting the bias for a set of events against the average forecast for the same set of events. We calculate bias as the mean
error





$$BIAS := \frac{1}{T} \sum_{t=1}^{T} \left( \bar{y}_{fcst,t} - y_{obs,t} \right) \tag{6}$$

where $\bar{y}_{fcst,t}$ is the forecast ensemble mean for event $t$ and $y_{obs,t}$ is the corresponding observation. Bias is calculated separately for each catchment, initialisation date and day. The bias is calculated across 23 events. For a given day, we calculate the

average absolute bias (AB) across all 12 catchments and 12 initialisation dates. Notwithstanding that the bias is scale dependent, the average absolute bias is used to compare the magnitude of biases for different model forecasts and for different days ahead. We do not compare relative biases because of the occasional divide by zero problem.

Reliability is the statistical consistency of forecasts and observations — a reliable forecasting system will accurately estimate the likelihood of an event. Reliability is checked by analysing the forecast probability integral transforms (PITs) of streamflow

observations. The PIT for a forecast-observation pair is defined by $\pi = F(y_{obs})$ where $F$ is the forecast cumulative distribution function (CDF). In the case that $y_{obs} = 0$, a pseudo-PIT value is sampled from a uniform distribution with a range $[0, \pi]$ (Wang and Robertson, 2011). If a forecasting system is reliable, $\pi$ follows a standard uniform distribution. Reliability can be visually examined by plotting the set of $\pi_t$ (t=1,2,…,T) with the corresponding theoretical quantile of the uniform distribution using the PIT uniform probability plot (or simply PIT plot. A perfectly reliable forecast follows the 1:1 line. In

this study, we do not plot individual PIT diagrams. Instead, reliability is summarised using the α-index (Renard et al., 2010)

$$\alpha = 1.0 - \frac{2}{n} \sum_{t=1}^{n} \left| \pi_t^* - \frac{t}{n+1} \right| \tag{7}$$

where $\pi_t^*$ is the sorted $\pi_t$ in increasing order. The α-index represents the total deviation of $\pi_t^*$ from the corresponding

uniform quantile (i.e. the tendency to deviate from the 1:1 line in PIT diagrams). The α-index ranges from 0 (worst reliability) to 1 (perfect reliability).

Forecast skill is evaluated using the continuous ranked probability score (CRPS; Matheson and Winkler, 1976). The CRPS for a given forecast and observation is defined as

$$CRPS = \int \left[ F(y) - H(y - y_{obs}) \right]^2 dy \tag{8}$$





where $y$ is the forecast variable; $y_{obs}$ is the observed value; $F$ is the forecast CDF ; and $H$ is the Heaviside step function, which equals 0 if $y < y_{obs}$ and equals 1 otherwise. Model forecasts are compared to reference forecasts by calculating skill scores:

$$\text{CRPS skill score} = \frac{\overline{\text{CRPS}_{\text{ref}}} - \overline{\text{CRPS}}}{\overline{\text{CRPS}_{\text{ref}}}} \times 100 \quad (\%) \tag{9}$$

where the overbar indicates averaging across a set of events. Reference forecasts are taken as the day-of-year climatology, fitted using data 5 days either side of the day-of-year, matching the window size that the BoM uses to post-process the QM forecasts (see section 3.4). The CRPS skill score is positively oriented (whereas CRPS is negatively oriented). As a percentage, a maximum score of 100 is indicative of perfect forecasts.  A score of 0 indicates no overall improvement compared to the reference forecast. A negative score indicates poor quality forecasts in the sense that a naïve climatology forecast is more
skilful.

### 3.4 Comparison with forecasts post-processed by using quantile mapping

We compare RPP-S forecasts with forecasts that have been post-processed at the ACCESS-S grid scale using quantile mapping (QM).  The QM forecasts are supplied by the Bureau of Meteorology alongside the raw forecasts. QM matches the statistical
distribution of past forecasts to the distribution of observations to reduce errors in the forecast mean and improve forecast spread (Crochemore et al., 2016;Zhao et al., 2017). A post-processed forecast value is obtained by first working out the quantile fraction (cumulative probability) of the new forecast using the CDF of past forecasts, then inverting the quantile fraction using the CDF of observations. The Bureau of Meteorology applied a separate quantile mapping model to each day. The CDF of the past forecasts is formed using 11 ensemble members in an 11-day sliding window and 22 years of data in leave-one-year-out
cross-validation. The statistical distribution of the observations is formed using the observations in an 11-day sliding window and 22 years of data. For the first 5 days the window is fixed to the first 11 days.  If the raw forecast ensemble member is above the previously known maximum forecast value, then the forecast value is instead linearly rescaled by $o_{\max} / f_{\max}$ where $o_{\max}$ is the maximum observed value and $f_{\max}$ is the maximum past forecast value.

### 4 Results

### 4.1 Bias in forecasts of daily amounts

Biases in forecasts of daily amounts are analysed for selected days using Figure 4. Each circle represents the bias for a catchment and initialisation date. The bias is plotted against the average forecast (averaged over all events). As expected, raw





forecasts are more biased than post-processed forecasts. The AB for raw forecasts ranges, for the examples given, from about 1.3mm to 1.5mm. The bias for raw forecasts tends to be negative, indicating that ACCESS-S has a propensity to underestimate daily rainfall amounts.

QM and RPP-S are similarly effective at reducing biases in daily amounts and both reduce bias significantly. After post-processing residual bias remains for any single day since the bias is corrected using pooled observations. The AB for QM forecasts ranges from about 0.8mm to 1.1mm. The AB for RPP-S forecasts ranges from about 0.6mm to 1.1mm. We do not think the differences in bias between QM and RPP-S are significant and the results vary for different selections of days (not shown). Visual examination of the QM and RPP-S scatter plots shows no wholesale bias in either the QM or RPP-S forecasts of daily amounts.

## 4.2. Bias in forecasts of accumulated totals

Biases in forecasts of accumulated totals are analysed using Figure 5 after rescaling to mm/day. As with the results for daily amounts, the raw forecasts are more biased than post-processed forecasts and the bias tends to be negative. Visual examination of the scatterplots for the raw forecasts reveals, particularly for days 30, 60 and 90, that for some catchments and initialisation times, the raw forecasts are unbiased for sub-seasonal to seasonal rainfall totals; which suggests that the GCM performs well in some regions. Nevertheless, it is evident that post-processing is still very necessary before using the ACCESS-S forecasts in hydrological forecasting.

QM and RPP-S both reduce bias significantly. In contrast to the results for daily amounts, QM and RPP-S have differing efficacy for reducing biases in accumulated totals. Visual examination of the QM and RPP-S scatter plots shows that the RPP-S points fit more tightly around the zero line than the QM points. The better ability of the RPP-S forecasts to reduce biases is reflected in the AB; for example, for day 90 the AB for QM forecasts is 0.28 mm/day and for RPP-S forecasts the AB is 0.12 mm/day (c.f. the AB for raw forecasts of 0.93 mm/day).

## 4.3 Reliability

Reliability is analysed using Figure 6, which presents boxplots of α-index for forecasts of daily rainfall amounts (left panel) and accumulated totals (right panel). The boxplots describe the distribution of α-index values for the same cases as we evaluated bias in section 4.2 except that we omit the day 1 result from the accumulated totals analysis. Results are presented for RPP-S before and after the Schaake Shuffle has been applied.

Raw forecasts have the poorest reliability for both daily amounts and accumulated totals. RPP-S has marginally better reliability than QM for daily amounts. It is noted that the Schaake Shuffle has no effect on forecast reliability for daily amounts since it is plainly a reordering of already randomised ensemble members over lead times.

RPP-S forecasts are more reliable than raw forecast and QM forecast for accumulated totals. The RPP-S forecasts become significantly more reliable after applying the Schaake Shuffle and linking the ensemble members into a realistic temporal pattern (discussed further in section 5).





## 4.4 Skill scores – overall performance

Skill scores for QM forecasts are plotted against skill scores for RPP-S forecasts in a scatterplot (Figure 7). Skill scores for daily amounts are plotted in the left panel; skill scores for accumulated totals are plotted in the right panel. Accumulated totals are for two days or more. RPP-S forecasts and QM forecasts tend to be positively skill for the same cases, although there is

considerable variation in the magnitude of the skill scores. A striking feature of the scatterplots is that when QM forecasts are negatively skilful, the RPP-S forecasts tend to be neutrally skilful, particularly for the accumulated totals. It is evident that skill for daily amounts can be sharply negative. The skill for daily amounts can be difficult to estimate because of the small sample size and the inability to accurately forecast daily amounts beyond about 10 days. In contrast, the skill for accumulated totals is easier to estimate as it benefits from temporal averaging and the bleeding through of skill from earlier periods.

## 4.5 Skill scores - detailed evaluation

Skill scores are partitioned according to catchment, days ahead and initialisation date in Figures 8, 9 and 10, respectively. Since forecasts of accumulated totals are more informative for water resources management and forecast skill is generally known to be limited beyond the first week or two, we focus the remainder of the results on accumulated totals.

CRPS skill scores are plotted for each catchment in Figure 8. Skill scores for RPP-S forecasts are vastly positive except for,

most noticeably, some cases in ORO, HRD and HLG. The negative skill scores for RPP-S forecasts in ORO, when skill scores from QM forecasts are positive, coincide with the very dry July-August period in that catchment (see ahead to Figure 10). It highlights an extreme case where there is insufficient information to fit the RPP-S model despite the pooling of data. In several catchments, e.g., WLC and HRD, RPP-S forecasts are seen to significantly outperform QM forecasts by virtue of QM forecasts being significantly negatively skilful and RPP-S forecasts rarely becoming negatively skilful. In several catchments, for

example, BRP and CTG, both QM and RPP-S skill scores are predominantly positive, demonstrating that simple bias correction techniques will appear sufficient in some cases.

CRPS skill scores are plotted for groups of days in Figure 9. For days 2–10, the RPP-S and QM forecasts are similarly skilful and vastly positively skilful. There are some instances of negative skill, which are likely to be artefacts of cross-validation. For days 11–19 and 20–28, the positive relationship between QM and RPP-S forecasts gradually weakens. From day 29

onwards, the QM forecasts become negatively skilful in many instances whereas the RPP-S skill scores tend to level out around zero and are rarely negative to the same degree. For days far ahead of model initialisation, QM skill scores can be higher than RPP-S skill scores although overall the skill scores for accumulated seasonal totals are quite low ($< 30\%$). Both these factors signal room for further improvement in RPP-S forecasts.

CRPS skill scores are plotted for each initialisation date in Figure 10. Forecasts initialised from 1 September to 1 November

have the highest proportion of cases where both QM and RPP-S forecasts are positively skilful, suggesting the ACCESS-S produces it's best sub-seasonal to seasonal forecasts during the Austral spring and summer.



## 5 Discussion

We demonstrate that RPP-S is able to improve daily GCM rainfall forecasts by: reducing bias (Figs. 4–5); improving reliability (Fig 6); and ensuring that forecasts are typically at least as skilful as a climatological reference forecast (Figs 7–10). RPP-S forecasts are comprehensively compared with quantile mapping (QM) forecasts. RPP-S forecasts outperform QM forecasts,

primarily because QM does not take into account the correlation between forecasts and observations. Since RPP-S is built upon the Bayesian joint probability (BJP) modelling approach, it explicitly models the correlation between the forecasts and observations, and thus takes into account model skill in the calibration. Our results add to the findings of Zhao et al. (2017) who studied the post-processing of monthly rainfall forecasts from POAMA (Australia's GCM preceding ACCESS-S). While Zhao et al. (2017) demonstrated that QM is very effective for bias correction, they did not consider accumulated totals. In our

study, we find that the RPP-S forecasts are less biased and more reliable than QM forecasts for accumulated totals.

Figure 5 illustrates the importance of the Schaake Shuffle for producing reliable forecasts using RPP-S. If the forecasts are not shuffled, then the forecasts of accumulated totals will tend to be too narrow in terms of ensemble spread, making the forecasts over-confident and reducing reliability. Related to this, RPP-S forecasts are more reliable for daily amounts than QM forecasts, yet QM and RPP-S forecasts do not exhibit any obvious differences in the magnitude of biases (Figure 3); evidence that the

RPP-S forecasts have a more appropriate ensemble spread, since the α-index integrates information about forecast bias and ensemble spread.

We establish six BJP models for each catchment and initialisation date to forecast 100+ days ahead. Alternative configurations are possible. For example, a separate BJP model can be established for each of the 100+ days where the data are pooled within a sliding window. The disadvantage of having a separate model for each day is the added computational cost. We are not

suggesting that it is infeasible (RPP-S is very fast), merely pointing out that the performance benefits would need to be weighed up against the computational costs and increase in the number of parameters in future investigations.

We applied a consistent methodology to perennial and ephemeral catchments, which is unlikely to be optimal. Our results for the ORO catchment (Figures 8–10) demonstrate that pooling seven days of data and two ACCESS-S runs is probably a minimum requirement in very dry catchments. In perennial catchments and, more specifically, where there are strong predictor-

predictand relationships, higher skill scores may be achieved by establishing additional models in the earlier periods. Conversely, in very dry periods it may be better to simply approximate the climatology well by pooling data over a longer period.

The RPP-S method is sophisticated in that it is a full calibration approach. However, there are opportunities to improve the methodology. For example, by pooling the data for many days in model parameter inference, it is assumed that rainfall from

one day to the next is independent, an assumption that is almost certainly false. New inference methods that treat the rainfall data as conditionally independent ought to be investigated. Another area to improve is to handle overlaps in data, which in our study arise as we pool a large number of days to approximate climatological distributions well. A simple strategy to avoid





overlaps in data would be to limit day groups to the size of the gap between GCM forecast initialisation dates. Future studies will seek to address the issues of independence and overlapping data in more sophisticated ways.

We make use of ACCESS-S runs initialised on day 1 of the month and day 25 of the previous month. These initialisation dates are only 4–7 days apart and therefore the climatology of daily rainfall is unlikely to change significantly over that period of time. It is technically possible to establish an RPP-S model using initialisation dates spanning several months. If far apart initialisation dates are included in model parameter inference, new strategies may be needed to ensure that the effects of seasonality are minimised. One possible approach is to standardise the forecasts and observations prior to fitting the BJP model. In this way, the model transformation and climatological parameters will be allowed to vary by day of year. Such strategies for building more robust RPP-S models and coincidentally minimising the effect of seasonality will be investigated in follow up work.

RPP-S and QM CRPS scores are calculated using ensembles of different sizes. When competing forecasting systems are perfectly reliable, larger ensembles should yield better CRPS scores (Ferro et al., 2008); however QM forecasts tend to be less reliable than RPP-S forecasts (Figure 5). It's our experience that narrow yet unreliable forecasts can score overly well in terms of CRPS. Despite these factors, our major conclusions are unlikely to change due to ensemble size effects.

RPP-S is designed to post-process forecasts of daily amounts. An alternative strategy is to post-process accumulated totals and subsequently disaggregate them to daily amounts. BJP models may be applied to post-process monthly and seasonal totals, which are then disaggregated them to daily time scales. Future work will investigate the relative merits of direct daily post-processing versus a seasonal calibration and disaggregation approach. This follow up work is of particular interest as our study has shown daily post-processing skill is limited beyond 10-15 days. It is not clear how much of the seasonal forecasting skill is attributable to skill in the initial period and how much of the skill is attributable to seasonal climate signals in the GCM.

An alternative to statistical post-processing of GCM outputs is to run a regional climate model (RCM) to provide much more localised information than a global GCM. A review study by (Xue et al., 2014) found that RCMs have limited downscaling ability for sub-seasonal to seasonal forecasts. In that regard, RCM outputs may be statistically post-processed also, which may lead to better forecasts in some regions. RCMs are suited to specialised studies and less suited to post-processing operational GCM forecasts in support of national scale hydrological forecasting services.

## 6 Conclusion

We have developed a novel method for post-processing daily rainfall amounts from seasonal forecasting GCMs. RPP-S is a full calibration approach that makes use of Bayesian joint probability (BJP) modelling to account for predictor-predictand skill relationships in the post-processing. Reliable forecasts of sub-seasonal and seasonal accumulated totals are produced by linking ensemble members together using the Schaake Shuffle.

We applied RPP-S to 12 catchments across Australia in diverse climate zones and to 12 ACCESS-S initialisation dates. The method is robust in terms of being capable of post-processing forecasts in all cases, even in very dry catchments.



Compared to raw forecasts and quantile mapping (QM) post-processing (which does not account for predictor-predictand skill relationships), RPP-S performs significantly better in terms of correcting bias, reliability and skill. The only exception to this conclusion is that QM and RPP-S are similarly effective for correcting biases in daily amounts. RPP-S is particularly effective at delivering reliable, skilful, monthly and seasonal rainfall forecasts. Thus RPP-S forecasts are highly suitable for feeding into

5   hydrological models for seasonal streamflow forecasting and other water resources management applications.

RPP-S is efficient and makes use only six calibration models to produce forecasts over 100 days ahead. Pooling multiple GCM runs and grouping forecast days in model parameter inference are practical measures applied to enable statistical post-processing across a range of perennial and ephemeral streams. There are many avenues of research that could significantly improve the robustness and performance of RPP-S forecasts.

10  **Acknowledgements**

This research was supported by the Water Information Research and Development Alliance (WIRADA), a partnership between CSIRO and the Bureau of Meteorology.  We thank the Bureau of Meteorology for providing the ACCESS-S data, AWAP data and catchment information used in this study. We thank Durga Lal Shrestha for helpful comments on the manuscript.





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



**Table 1 Catchment ID, catchment name (river and gauging location), gauge ID, region and catchment size**

| ID | Catchment name | Gauge id | Region | Area (km²) |
|----|----------------|----------|--------|-----------|
| BRP | Barron River above Picnic Crossing | 110003A | Queensland | 228 |
| BRS | Burdekin River above Sellheim | 120002 | Queensland | 36260 |
| DIB | Diamantina River at Birdsville | A0020101 | Queensland | 119034 |
| NMN | Namoi River above North Cuerindi | 419005 | New South Wales | 2532 |
| WLC | Wollomombi River above Coninside | 206014 | New South Wales | 377 |
| CTG | Cotter River above Gingera | 410730 | Murray-Darling Basin | 148 |
| MRB | Murray River above Biggara | 401012 | Murray-Darling Basin | 1165 |
| DVC | Davey River D/S Crossing River | 473 | Tasmania | 698 |
| HLG | Hellyer River above Guilford Junction | 61 | Tasmania | 100 |
| DRT | Deep River above Teds Pool | 606001 | Western Australia | 474 |
| HRD | Harvey River above Dingo Road | 613002 | Western Australia | 148 |
| ORO | Ord River at Old Ord Homestead | 809316 | Northern Australia | 19513 |



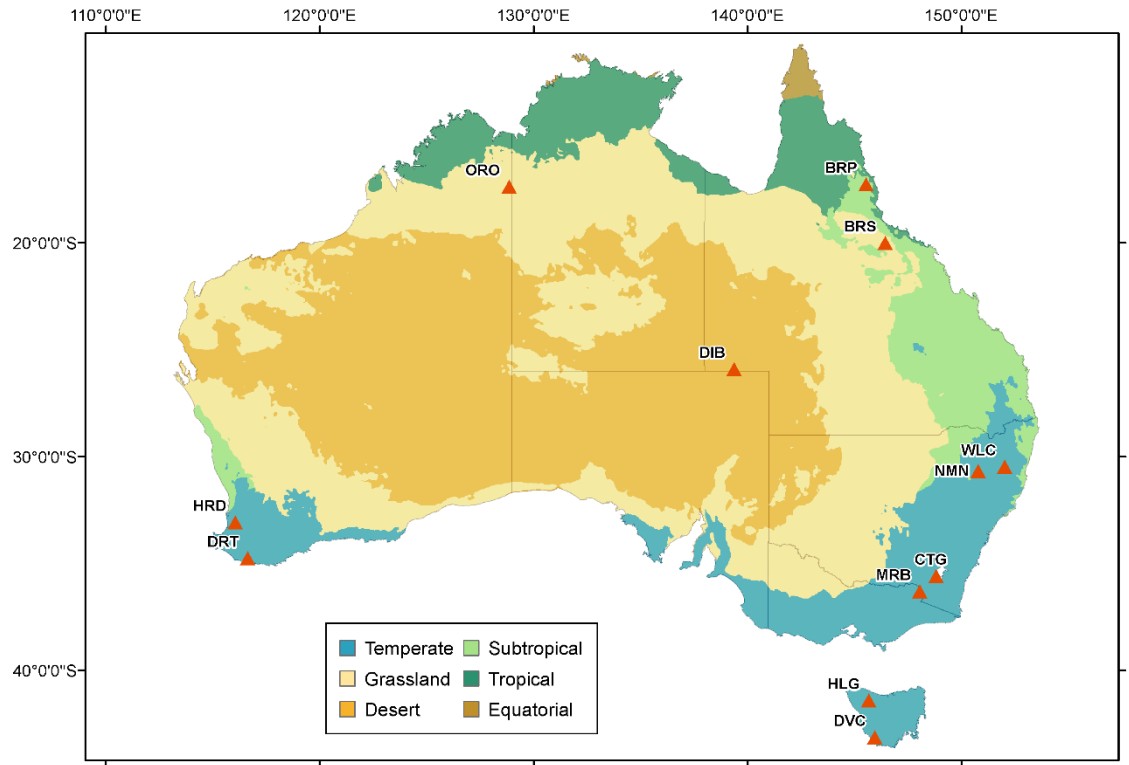

**Figure 1. Map of Australian climate zones overlaid by gauging locations plotted as red triangles and labelled with catchment ID. Details of the catchments, including size, are presented in Table 1.**





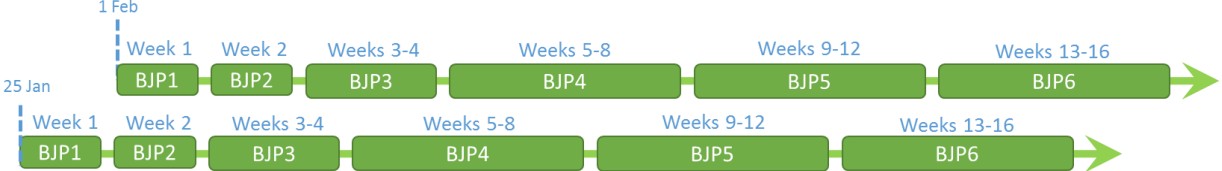

**Figure 2. Schematic of data used to establish six BJP models for calibrating daily rainfall forecasts initialised on 1 February. Data are pooled for contiguous numbers of days since forecast. Each BJP model is used to calibrate daily forecasts within the time period it covers.**

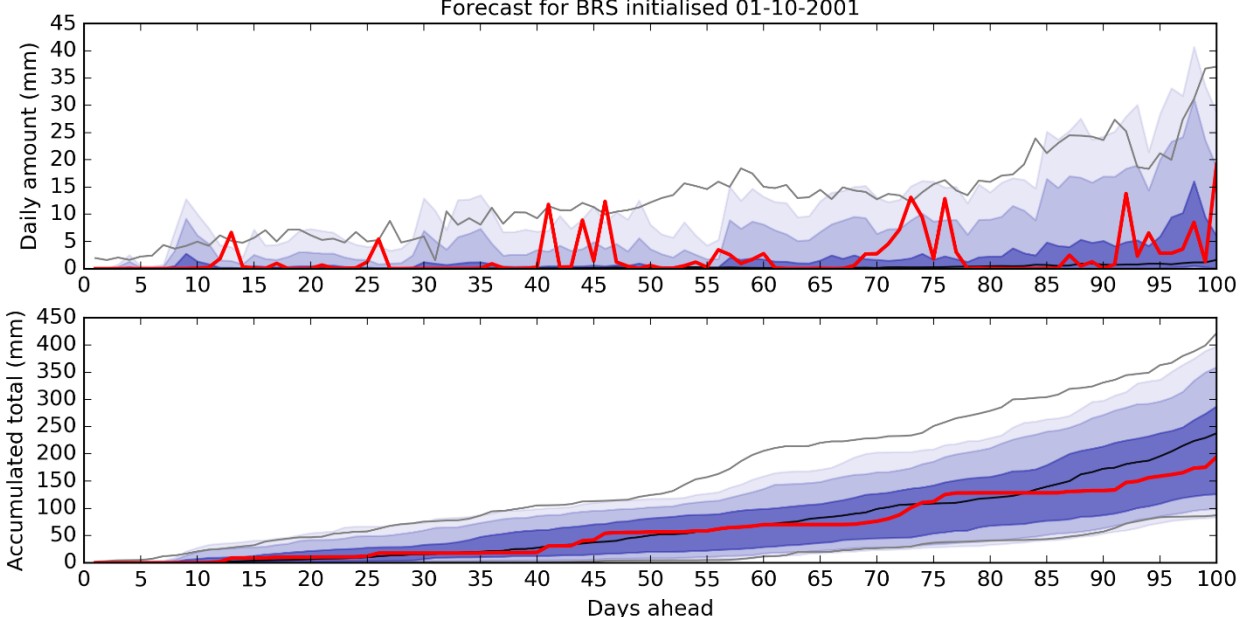

**Figure 3. Example RPP-S rainfall forecast for the Burdekin River at Sellheim, initialised on the 1 October 2001 and forecasting 100 days ahead. Forecasts of daily amounts are shown in the top panel and forecasts of accumulated totals are shown in the bottom panel. Dark blue is the forecast [0.25,0.75] quantile range, medium blue is the forecast [0.10,0.90] quantile range and light blue is the forecast [0.05, 0.95] quantile range. Grey lines are the climatological reference forecast [0.05, 0.95] quantile range. The black line is the climatological reference forecast median. The red line is the observation.**



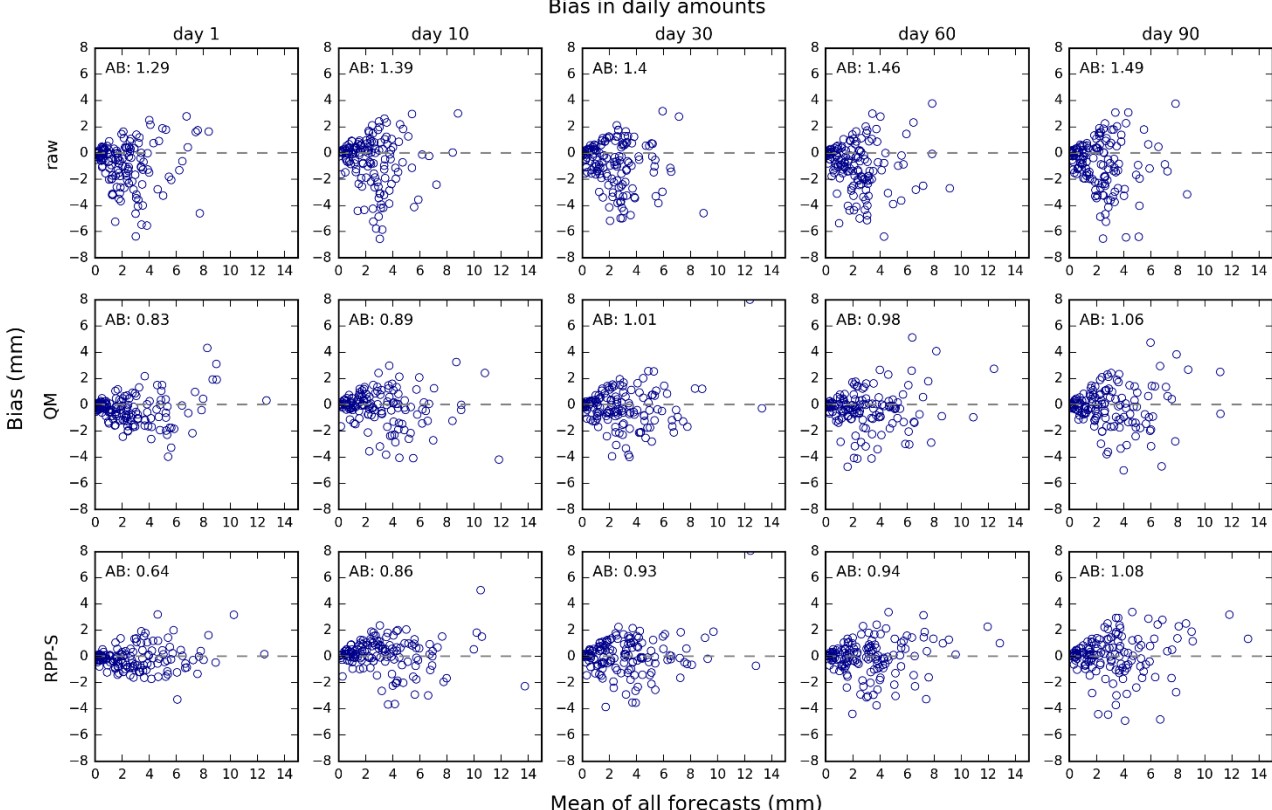

**Figure 4.** Bias in daily rainfall forecasts for raw, QM and RPP forecasts (rows) and selected days ahead (columns). The scatterplots are of forecast mean (x-axis) versus observation (y-axis). Both axes are in mm. There is one blue circle for each catchment and forecast initialisation time. The average absolute bias (AB) is printed in the top left corner.




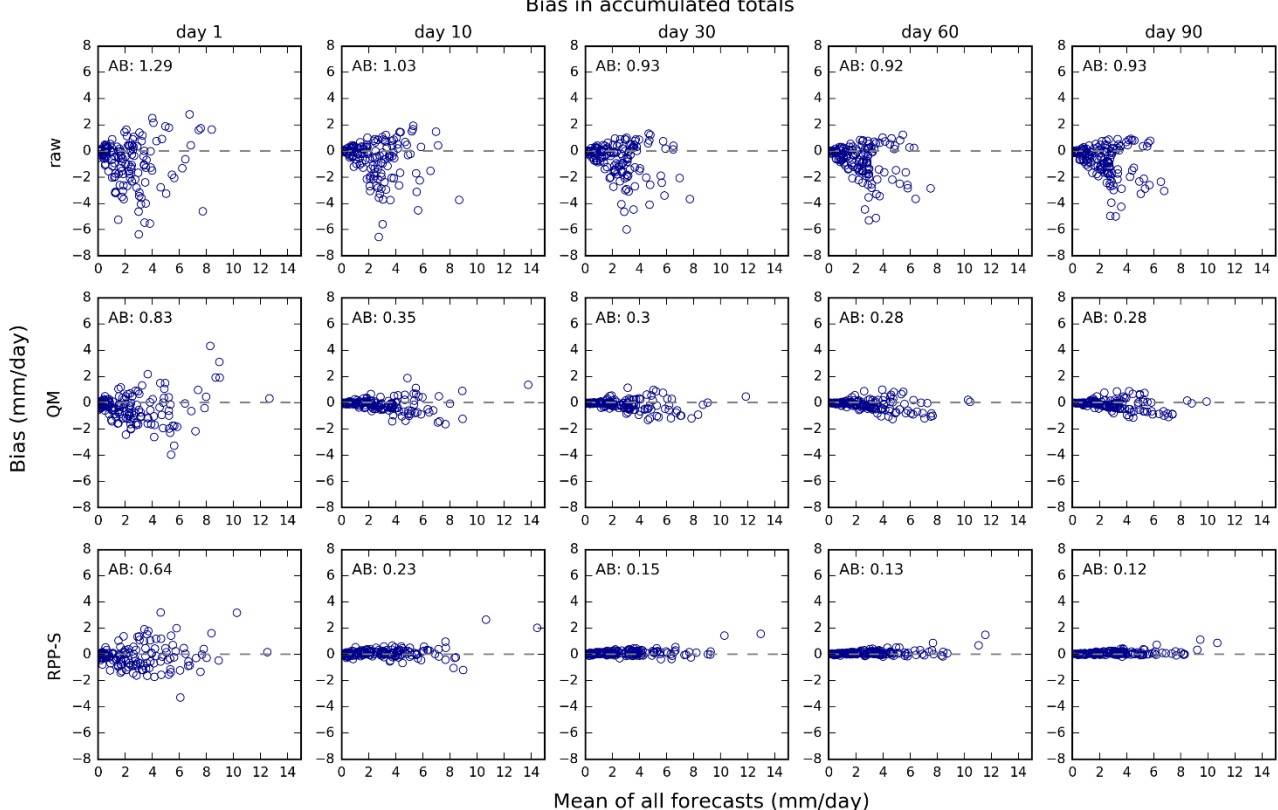

**Figure 5. As for Figure 4, except for accumulated totals**





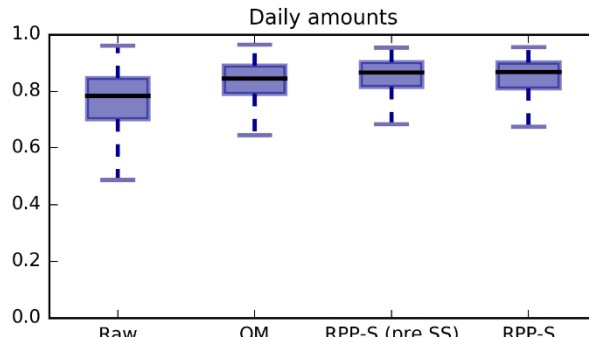 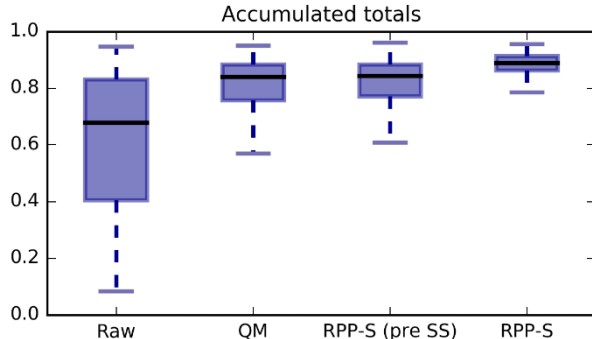

**Figure 6. α-index of reliability for forecast daily rainfall amounts (left panel) and forecast accumulated rainfall totals (right panel). Results for four types of forecasts are presented: raw, QM, RPP-S before Schaake Shuffle (pre SS) and RPP-S forecasts after the Schaake Shuffle. Higher α-index indicates better reliability. The boxplots display the median as a black line. The box spans the interquartile range and the whiskers span the [0.1,0.9] quantile range.**




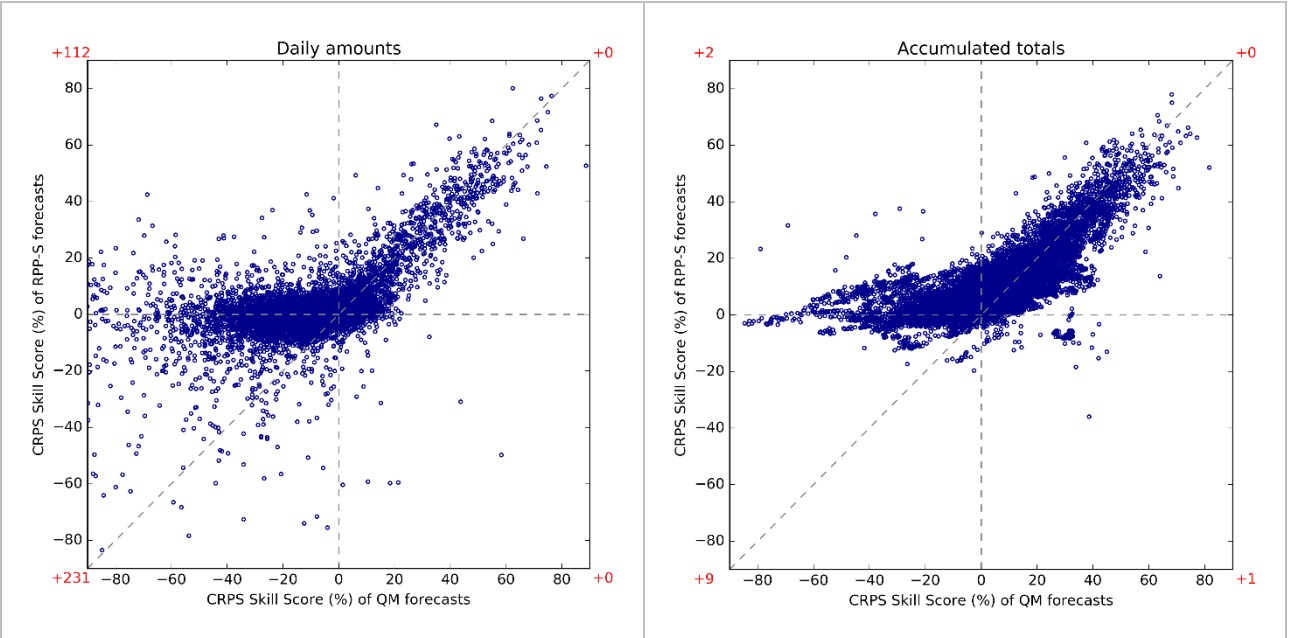

**Figure 7. Scatterplots of CRPS skill scores for daily amounts (left panel) and accumulated totals (right panel). Results for QM are on the horizontal axis and results for RPP-S are on the vertical axis. Higher CRPS skills scores reflect better forecast performance. Red text preceded by a "+" symbol indicates the number of points plotted outside the axis limits in the quadrant nearest the text.**




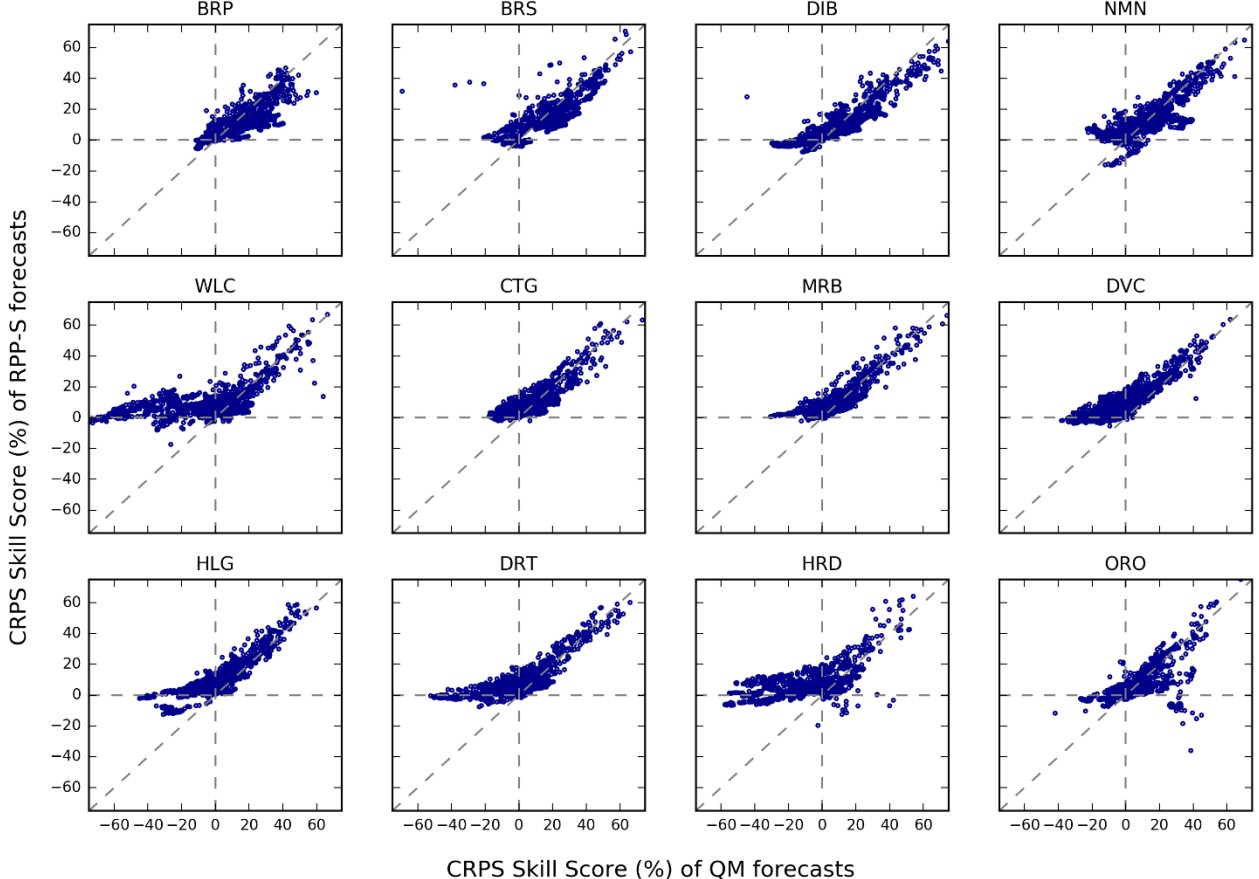

**Figure 8. Scatterplots of CRPS skill scores for accumulated totals for each catchment. The scatterplots plot QM skill scores (horizontal axis) against RPP-S skill scores (vertical axis). Each scatterplot pools results for all initialisation dates and all days ahead.**





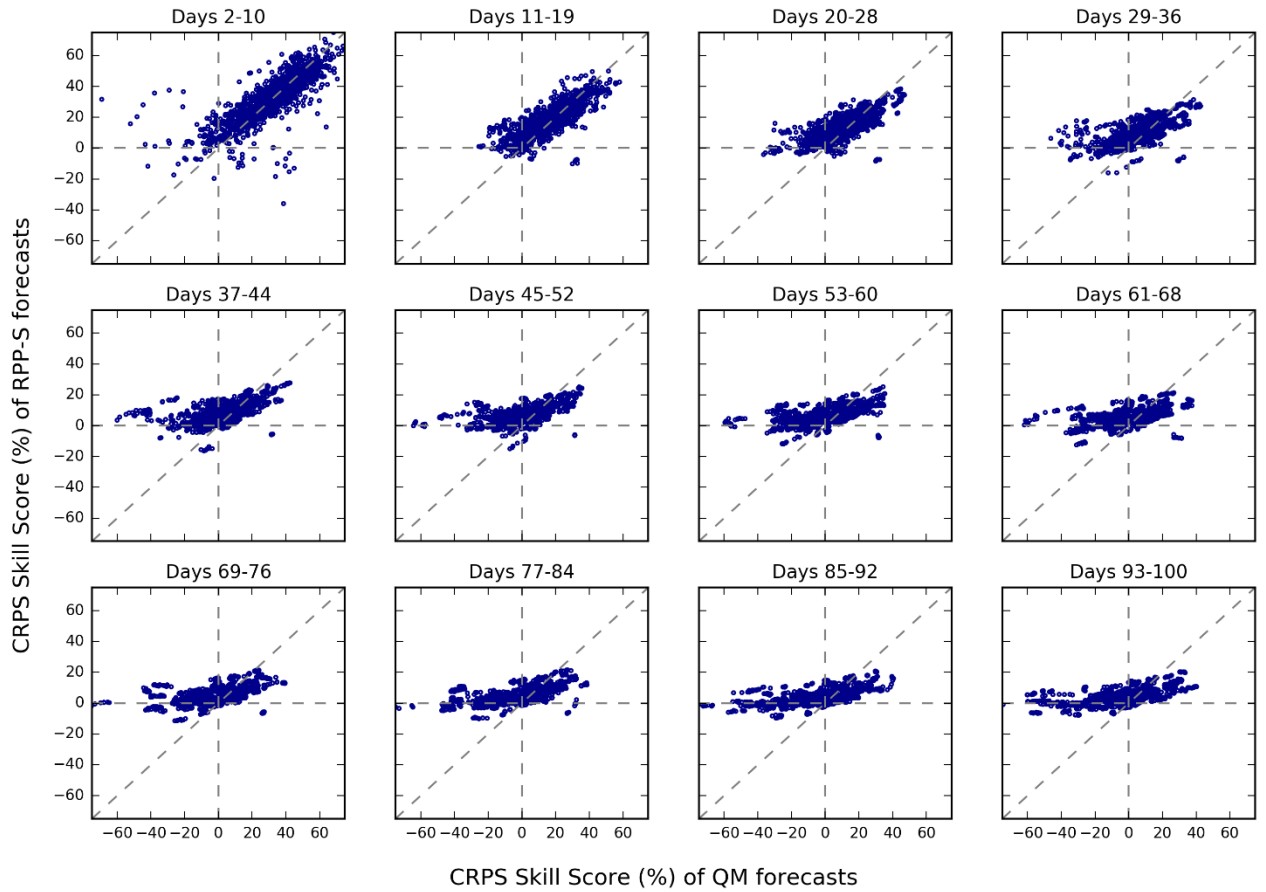

**Figure 9. Scatterplots of CRPS skill scores for groups of days. The scatterplots plot QM skill scores (horizontal axis) against RPP-S skill scores (vertical axis). Each scatterplot pools results for all initialisation dates and all catchments.**





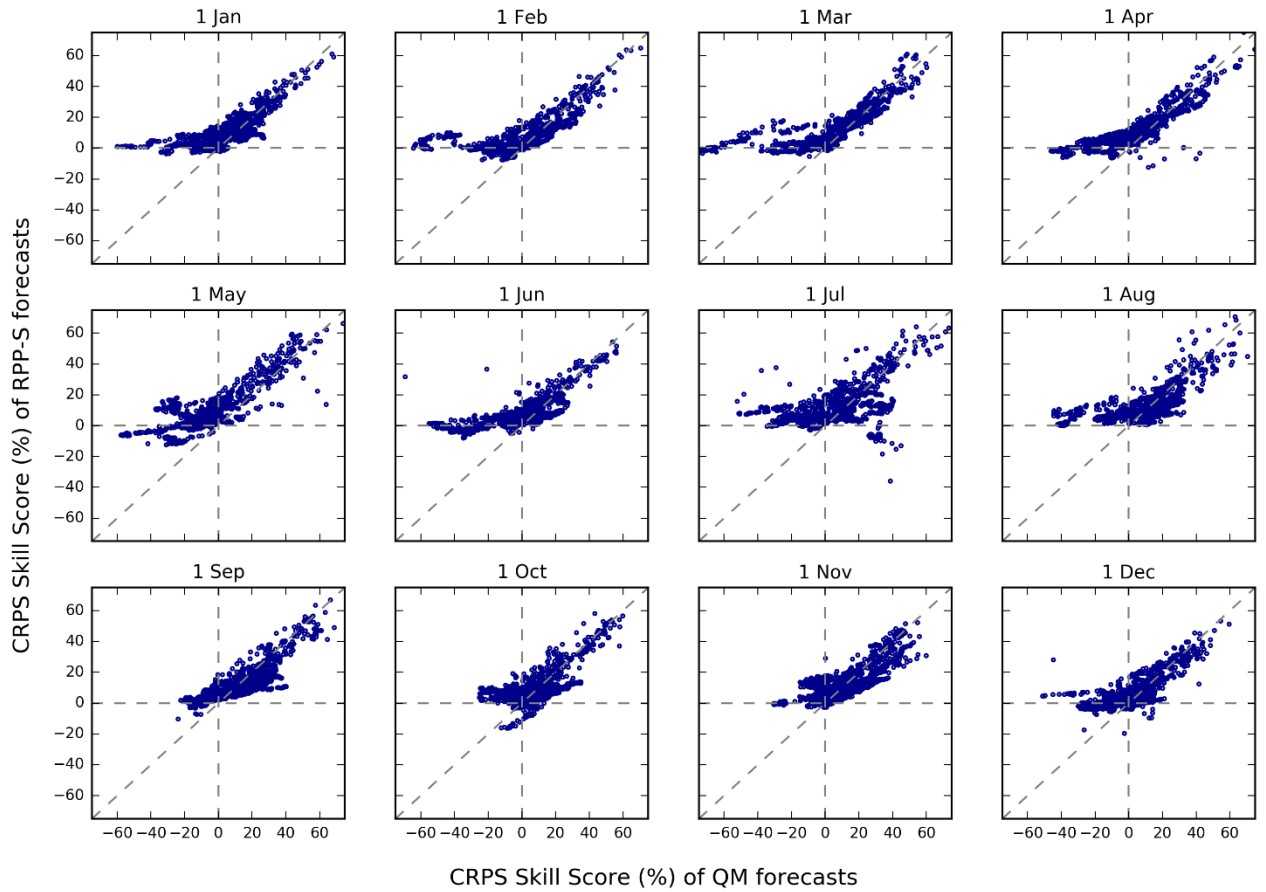

**Figure 10. Scatterplots of CRPS skill scores for accumulated totals for each initialisation date. The scatterplots plot QM skill scores (horizontal axis) against RPP-S skill scores (vertical axis). Each scatterplot pools results for all catchments and all days ahead.**

