# Peer review of "A Bayesian modelling method for post-processing daily sub-seasonal to seasonal rainfall forecasts from GCMs and evaluation for 12 Australian catchments"

_Hydrology and Earth System Sciences, 2017_

## Referee Comment (RC1) · Anonymous Referee #1 · 7 Aug 2017

General comments The paper addresses the important issue of calibrating outputs from a General Circulation Model (GCM) for 12 Australian catchments. The authors implement two relevant methodological choices: (i) calibration of daily precipitation and (ii) use of a calibration method that accounts for the correlation between forecasts and observations. Furthermore, the authors evaluate the added value (value as a gain in skill) that the proposed method has over the use of raw forecasts and the simpler and popular Quantile Mapping (QM). The paper builds upon the work of the research group which fits nicely within the general scope of HESS and within the objectives of the "Sub-seasonal to seasonal hydrological forecasting" special edition. I believe that post-processing of seasonal forecasts is a subject still in its infancy. Any effort to

improve the forecasts is a relevant addition to the field of forecasting due to the negative societal effects of issuing forecasts that exhibit biases. My recommendation is that the paper should be accepted after minor revisions which are mainly clarifications needed to ease the reading. Detailed comments are given below.

Specific comments

(Major Comments)

Page 3, Lines 30-32 and Sect. 3.4: As I understand it, the set up for the QM uses a sliding window of 11-days in order to estimate the predictive and the observed distribution functions, while RPP-S uses 14-days for week 1 and week 2 up to 56 days for weeks 13-16. I think the implementation of both methods should be consistent in order to achieve robust conclusions. Perhaps something in the lines of what is discussed in Page 12, Lines 17-21. Could the authors explain how the difference in implementations between QM and RPP-S may impact the conclusions?.

Page 5, Line 18: To my understanding, the number of ensembles after post-processing is 1000, is that correct? If this is the case, then I would consider the comparison to the raw and quantile mapping corrected forecasts unfair. Perhaps the authors should make an effort to debias the CRPS as in Ferro, et al., (2008) to add quantitative evidence of the claim in Page 13 Lines 11-14. On the other hand, if the number of ensembles is 11, then it should be stated more clearly on the text.

(Minor comments)

Page 3, Lines 30-32: These lines can be removed as you are also explaining this on Sect. 3.4.

Page 4-7: Merge Sect. 3.1 with Sect. 3.2.2 and 3.2.3 to avoid repetition and to make the manuscript shorter.

Page 7, Lines 14-20. Perhaps change this paragraph to the result section.

Page 7, Lines 26-27. These two lines should be moved somewhere in Sect. 3.2.

Page 8, Line 1. Change BIAS for AB in Eq. (8).

Page 8, Lines 2-4 Repetitive sentence, I suggest it is changed from "Bias is calculated separately for each catchment, initialization date and day. The bias is calculated across 23 events. For a given day, we calculate the average absolute bias across all 12 catchments and 12 initialization dates." to "Bias is calculated separately for each of the 12 catchments, 12 initialization dates and day across the 23 events.".

Page 10, Line 11. Could the authors clarify what do they mean by "after rescaling to mm/day"?.

Page 11, Sect. 4.4 and 4.5. I understand how the shuffling affects the reliability of the accumulated totals as discussed in Sect 4.3. However it is not clear if in the evaluation of the skill scores you are using the shuffled RPP-S. Could you clarify this matter? Also, it is not clear how the accumulation is done. Are you accumulating rainfall from the start of the forecast until 2 days after, then 3 days, and so on until 112 days?. Then, in total you have 12 (catchments) x 12 (initialization days) x 111 (accumulation periods) number of points in Fig. 7 right panel (and similarly 12x12x112 on the left panel)? This is my interpretation of Page 11, Line 3-4 that reads "Accumulated totals are for two days or more. ...". Further explanation on the accumulation is also needed for Fig. 8 to Fig. 10.

Page 12, Line 13: I suggest changing "... RPP-S forecasts are more reliable for daily amounts than QM forecasts" to "... RPP-S forecasts are slightly more reliable for daily amounts than QM forecasts".

Page 17, Table 1: Order catchments from smaller to bigger to make the visualization easier.

Page 18, Fig. 1: Add catchment boundaries to map in figure to help the reader relate Fig. 8 to catchment size.

Page 21 and 22, Fig. 4 and Fig. 5: Change label "Bias (mm)" to "AB (mm/day)".

Technical Corrections

Page 4, Line 14. "are mapped in Figure 1" should be "are mapped in Fig. 1". In general, mentions to figures should be changed throughout the manuscript to comply with HESS guidelines (https://www.hydrology-and-earth-system-sciences.net/for_authors/manuscript_preparation.html).

Page 5, Line 1. Missing parenthesis at the end of Eq. (1).

Page 8, Line 9. Change "streamflow" to "rainfall".

Page 8, Line 14. Missing parenthesis after "... (or simply PIT plot) ... ".

Page 11, Line 9. I am not sure the word "bleeding" is the correct one.

Page 13, Line 13. Change "(Figure 5)" to "(Fig. 6)".

References

Ferro, C. A. T., Richardson, D. S., and Weigel, A. P.: On the effect of ensemble size on the discrete and continuous ranked probability scores, Meteorol. Appl., 15, 19-24, 2008.

---

## Referee Comment (RC2) · Anonymous Referee #2 · 8 Aug 2017

General comments: This manuscript reports the development of a rainfall post-processor for GCM forecasts in the sub-seasonal to seasonal period (RPP-S). The proposed method is surely an important contribution as it attempts to advance in methods for post-processing rainfall forecasts in this time scale. The method elaborates on authors' previous work and makes use of the Bayesian joint probability (BJP) modeling approach to account for predictor-predictand skill relationships. The post-processor generates daily amounts which are then aggregated to in-season totals using the Schaake Shuffle. The method is applied to rainfall forecasts from the ACCESS-S model for a set of catchments in Australia, and is found more skillful than ACCESS-S forecasts post-processed using quantile mapping (QM).

[Figure]

I find the paper well written and the experimental setting well described, although in some instances additional clarifications would be desirable. I have only minor comments on some methodological assumptions which need more justification to ease the readability and warrant reproducibility of the proposed method. Apart from that, I found this manuscript suitable to be published on this special issue. Below, I elaborate these minor/specific comments:

Specific comments: In section 3.2.1, pooling of multiple GCM runs and grouping forecast days are key steps in the proposed method. The proposed RPP-S follows a particular configuration and authors argue that this is a practical measure to enable post-processing of rainfall forecasts across a range of perennial and ephemeral catchments, but there is no restriction for the RPP-S configuration. P6 L11-L13: "….The size of the day groups expands farther from the initialization day. Smaller day groups in early periods are intended to extract skill from initial conditions. Larger day groups in later periods are intended to better approximate the climatological distributions." Could the authors elaborate this statement in a more generic way? The point being, how can one estimate the size of day groups, without knowledge of the rainfall forecasts distribution properties? I see this issue is extensively discussed later in section 5, e.g. lines: L22-L27 in P12. Perhaps, some elements of the discussion should come earlier in the paper, e.g. in section 3.2.1

P7 L10-L13: "BJP forecast ensemble members are randomized and are not linked across days by default. To deal with the problem, we apply the Shaake Shuffle…." The use of the Shaake Shuffle approach is an important component of the proposed method, which is of major relevance when looking at time windows beyond the weather scale. Given the reliance on such technique to create realistic temporal patterns from BJP forecasts, a few lines describing details of the Shaake Shuffle rationale and its operational implementation are needed.

P12 L4-L5: "RPP-S forecast outperform QM forecasts, primarily because QM does not take into account the correlation between forecast and observations..". I also share

the first reviewer's concerns on how differences in the implementation of the QM (as described in section 3.4) and the RPP-S method could impact the results from the benchmarking experiment. It would be worth to discuss at length those differences and their implications on the conclusions.

Technical corrections:

P3 L25: Reference Hudson et al. (2017) is missing in the list of references

P5 L11: Symbols in Eq. (5) are no described

P8 L9: "….the forecast probability integral transforms (PITs) of streamflow observations. . ." Is it not rainfall observations?

P8 L14: ". . .probability plot (or simply PIT plot." Missing parenthesis

P12 L14: ". . .yet QM and RPP-S forecasts do not exhibit any obvious differences in the magnitude of biases (Figure 3)". Is it not Figure 4?

P13 L13: "reliable than RPP-S forecasts (Figure 5)". Is it not Figure 6?

---

## Author Comment (AC1) · 15 Sep 2017

**Review #1**

*General comments*

The paper addresses the important issue of calibrating outputs from a General Circulation Model (GCM) for 12 Australian catchments. The authors implement two relevant methodological choices: (i) calibration of daily precipitation and (ii) use of a calibration method that accounts for the correlation between forecasts and observations. Furthermore, the authors evaluate the added value (value as a gain in skill) that the proposed method has over the use of raw forecasts and the simpler and popular Quantile Mapping (QM). The paper builds upon the work of the research group which fits nicely within the general scope of HESS and within the objectives of the "Sub-seasonal to seasonal hydrological forecasting" special edition. I believe that post-processing of seasonal forecasts is a subject still in its infancy. Any effort to improve the forecasts is a relevant addition to the field of forecasting due to the negative societal effects of issuing forecasts that exhibit biases. My recommendation is that the paper should be accepted after minor revisions which are mainly clarifications needed to ease the reading. Detailed comments are given below.

Response: Thank you for the positive comments. We hope that this work encourages new and innovative approaches to calibration of seasonal forecasts and delivers better forecasts for end users.

*Specific comments — Major*

Page 3, Lines 30-32 and Sect. 3.4: As I understand it, the set up for the QM uses a sliding window of 11-days in order to estimate the predictive and the observed distribution functions, while RPP-S uses 14-days for week 1 and week 2 up to 56 days for weeks 13-16. I think the implementation of both methods should be consistent in order to achieve robust conclusions. Perhaps something in the lines of what is discussed in Page 12, Lines 17-21. Could the authors explain how the difference in implementations between QM and RPP-S may impact the conclusions?

Response: It is entirely reasonable to suggest that the RPP-S and QM models should make use of similar data windows to give more confidence in the conclusions. Therefore, we will modify the manuscript and show results with RPP-S set up to use an 11-day sliding window. That is, a separate RPP-S model will be used to forecast each day, in the same way as a QM model. Our conclusions will not change materially. In fact, we will see improvements in the results for some cases like the dry ORO catchment, presumably due to the inclusion of more information in the early forecasts.

Page 5, Line 18: To my understanding, the number of ensembles after post-processing is 1000, is that correct? If this is the case, then I would consider the comparison to the raw and quantile mapping corrected forecasts unfair. Perhaps the authors should make an effort to debias the CRPS as in Ferro, et al., (2008) to add quantitative evidence of the claim in Page 13 Lines 11-14. On the other hand, if the number of ensembles is 11, then it should be stated more clearly on the text.

Response: The RPP-S forecasts are made up of 1000 ensemble members; correct. We considered carefully the impact of the ensemble sizes during the course of the study and indeed did consider adjusting the CRPS following the work of Ferro et al.; however, the adjustment of Ferro et al. applies in an idealised situation and from our point of view isn't relatable if forecasts from different models exhibit differences in reliability. Fricker et al. (2013) describe how CRPS is an unfair measure in the sense that it discourages forecasting extremes, which is consistent with our experience that ensembles that are unbiased but too narrow (like the QM forecasts) can be scored overly well by CRPS. Since the QM forecasts are not similarly reliable to the RPP-S forecasts, the QM forecasts tend

to be too narrow, and we have not attempted to sharpen the RPP-S forecasts in order to improve their CRPS skill, we think it is difficult to make meaningful adjustments. We prefer to discuss the issue as we have in section 5. In our revision we will endeavour to explain the issue more satisfactorily in line with this response.

*Specific comments — Minor*

Page 3, Lines 30-32: These lines can be removed as you are also explaining this on Sect. 3.4.

Response: Agreed, we will remove the lines here.

Page 4-7: Merge Sect. 3.1 with Sect. 3.2.2 and 3.2.3 to avoid repetition and to make the manuscript shorter.

Response: We will streamline sections 3.1 and 3.2.

Page 7, Lines 14-20. Perhaps change this paragraph to the result section.

Response: During the manuscript preparation we considered including the forecast example in the results but we feel it sits better in the methods section.

Page 21 and 22, Fig. 4 and Fig. 5: Change label "Bias (mm)" to "AB (mm/day)".

Response: We have defined bias in equation (6) and therefore the y-axis label is correct as Bias (mm).

*Technical Corrections*

Page 4, Line 14. "are mapped in Figure 1" should be "are mapped in

Fig. 1". In general, mentions to figures should be changed throughout the

manuscript to comply with HESS guidelines (https://www.hydrology-and-earth-systemsciences.

net/for_authors/manuscript_preparation.html).

Page 5, Line 1. Missing parenthesis at the end of Eq. (1).

Page 8, Line 9. Change "streamflow" to "rainfall".

Page 8, Line 14. Missing parenthesis after "... (or simply PIT plot) ... ".

Page 11, Line 9. I am not sure the word "bleeding" is the correct one.

Page 13, Line 13. Change "(Figure 5)" to "(Fig. 6)".

Response: Thank you for reading our manuscript so closely – these are all changes we will now address.

*General comments*

This manuscript reports the development of a rainfall postprocessor for GCM forecasts in the sub-seasonal to seasonal period (RPP-S). The proposed method is surely an important contribution as it attempts to advance in methods for post-processing rainfall forecasts in this time scale. The method elaborates on authors' previous work and makes use of the Bayesian joint probability (BJP) modelling approach to account for predictor-predictand skill relationships. The post-processor generates daily amounts which are then aggregated to in-season totals using the Schaake Shuffle. The method is applied to rainfall forecasts from the ACCESS-S model for a set of catchments in Australia, and is found more skillful than ACCESS-S forecasts post-processed using quantile mapping (QM).

I find the paper well written and the experimental setting well described, although in some instances additional clarifications would be desirable. I have only minor comments on some methodological assumptions which need more justification to ease the readability and warrant reproducibility of the proposed method. Apart from that, I found this manuscript suitable to be published on this special issue. Below, I elaborate these minor/specific comments:

Response: Thank you for the positive comments and understanding of our work.

*Specific comments*

In section 3.2.1, pooling of multiple GCM runs and grouping forecast days are key steps in the proposed method. The proposed RPP-S follows a particular configuration and authors argue that this is a practical measure to enable post-processing of rainfall forecasts across a range of perennial and ephemeral catchments, but there is no restriction for the RPP-S configuration. P6 L11-L13. The size of the day groups expands farther from the initialization day. Smaller day groups in early periods are intended to extract skill from initial conditions. Larger day groups in later periods are intended to better approximate the climatological distributions." Could the authors elaborate this statement in a more generic way? The point being, how can one estimate the size of day groups, without knowledge of the rainfall forecasts distribution properties? I see this issue is extensively discussed later in section 5, e.g. lines: L22-L27 in P12. Perhaps, some elements of the discussion should come earlier in the paper, e.g. in section 3.2.1

P12 L4-L5: "RPP-S forecast outperform QM forecasts, primarily because QM does not take into account the correlation between forecast and observations.". I also share the first reviewer's concerns on how differences in the implementation of the QM (as described in section 3.4) and the RPP-S method could impact the results from the benchmarking experiment. It would be worth to discuss at length those differences and their implications on the conclusions.

Response: We respond to the above two comments together since they are related. It is right that we intend for the RPP-S approach to be flexible in terms of how forecasts and observations are paired and how days are grouped together. The best way to do it will depend on the available GCM data as well as catchment characteristics. In our study we apply a consistent method to all catchments and therefore we are comfortable with having a deferred discussion about other possible ways to set up models.

Regarding the concern about differences in model set up, we wish to give confidence in our method and results. Therefore, we will modify the manuscript and show results with RPP-S set up to use an 11-day sliding window. That is, a separate RPP-S model will be used to forecast each day, in the

same way as a QM model. Our conclusions will not change materially. In fact, we will see improvements in the results for some cases like the dry ORO catchment, presumably due to the inclusion of more information in the early forecasts.

P7 L10-L13: "BJP forecast ensemble members are randomized and are not linked across days by default. To deal with the problem, we apply the Shaake Shuffle."The use of the Shaake Shuffle approach is an important component of the proposed method, which is of major relevance when looking at time windows beyond the weather scale. Given the reliance on such technique to create realistic temporal patterns from BJP forecasts, a few lines describing details of the Shaake Shuffle rationale and its operational implementation are needed.

Response: We agree that a fuller description of the Schaake Shuffle is warranted and we will include such a description in our revised manuscript.

*Technical corrections*

P3 L25: Reference Hudson et al. (2017) is missing in the list of references

P5 L11: Symbols in Eq. (5) are no described

P8 L9: "the forecast probability integral transforms (PITs) of streamflow observations: : :" Is it not rainfall observations?

P8 L14: "probability plot (or simply PIT plot." Missing parenthesis

P12 L14: "yet QM and RPP-S forecasts do not exhibit any obvious differences in the magnitude of biases (Figure 3)". Is it not Figure 4?

P13 L13: "reliable than RPP-S forecasts (Figure 5)". Is it not Figure 6?

Response: Thank you for reading our manuscript so closely – these are all changes we will now address.

**References**

Fricker, T. E., Ferro, C. A. T. and Stephenson, D. B. (2013), Three recommendations for evaluating climate predictions. Met. Apps, 20: 246–255. doi:10.1002/met.1409

---

## Author Comment (AC2) · 15 Sep 2017

Please see AC1 and supplement
* * *

---

## Author Comment (AC3) · 15 Sep 2017

Please see AC1 and supplement
* * *